# Identification of Breathing Patterns through EEG Signal Analysis Using Machine Learning

**DOI:** 10.3390/brainsci11030293

**Published:** 2021-02-26

**Authors:** Yong-Gi Hong, Hang-Keun Kim, Young-Don Son, Chang-Ki Kang

**Affiliations:** 1Department of Health Sciences and Technology, Gachon Advanced Institute for Health Sciences & Technology, Gachon University, Incheon 21936, Korea; hyg4438@bme.gachon.ac.kr (Y.-G.H.); ydson@bme.gachon.ac.kr (Y.-D.S.); 2Department of Biomedical Engineering, Gachon University, Incheon 21936, Korea; 3Department of Radiological Science, Gachon University, Incheon 21936, Korea

**Keywords:** EEG, breathing, machine learning, LDA, random forest, working memory task

## Abstract

This study was to investigate the changes in brain function due to lack of oxygen (O_2_) caused by mouth breathing, and to suggest a method to alleviate the side effects of mouth breathing on brain function through an additional O_2_ supply. For this purpose, we classified the breathing patterns according to EEG signals using a machine learning technique and proposed a method to reduce the side effects of mouth breathing on brain function. Twenty subjects participated in this study, and each subject performed three different breathings: nose and mouth breathing and mouth breathing with O_2_ supply during a working memory task. The results showed that nose breathing guarantees normal O_2_ supply to the brain, but mouth breathing interrupts the O_2_ supply to the brain. Therefore, this comparative study of EEG signals using machine learning showed that one of the most important elements distinguishing the effects of mouth and nose breathing on brain function was the difference in O_2_ supply. These findings have important implications for the workplace environment, suggesting that special care is required for employees who work long hours in confined spaces such as public transport, and that a sufficient O_2_ supply is needed in the workplace for working efficiency.

## 1. Introduction

Breathing plays an important role in the vital activities of a person and affects many areas, such as learning ability, cognitive ability, health, and sleep. Therefore, it is very important to understand and study breathing from an ergonomic perspective. Breathing methods are primarily divided into mouth and nose breathing. In some studies, mouth breathing has been known to be associated with side effects, such as facial shape deformation [1] and decreased learning ability [2,3], when compared to nose breathing [1,4,5]. Furthermore, when adolescents breathe through their mouths, they have substantial difficulty exercising working memory and arithmetic skills due to insufficient O_2_ supply to the brain compared to nose breathing [2,3].

It has been reported that breathing methods can affect brain functions in working memory [2], but most existing research has found that the effects of breathing methods on brain functions were indirect and were expected to take place over a long period of time. For example, mouth breathing may be related to the learning ability of children, but its effect may not be direct and immediate. Furthermore, sleep disorders possibly caused by mouth breathing are also considered to affect learning ability [2]. However, recent studies have attempted to confirm the direct and immediate effects of breathing methods on brain function, and one of them confirmed the effects of breathing methods on working memory through electroencephalography (EEG) data [6]. The study reported that mouth breathing decreased the ability of working memory compared to nose breathing, suggesting that the biggest difference between mouth and nose breathing may be the O_2_ supply to the brain.

The performance of the brain is very sensitive to a reduced O_2_ supply, and it has already been confirmed that the performance for single- and multi-tasking was improved in a work environment with increased O_2_ supply to the brain [7]. In other words, O_2_ supply is very important to improve work efficiency and brain performance. Studies have also reported that the administration of high concentrations of O_2_ could lead to improvement in cognitive ability, such as a decrease in the reaction time during the 1-back task [8]. Therefore, it can be stated that an adequate supply of O_2_ in the working environment has a great influence on brain function.

In a recent study that investigated the direct and immediate effects of breathing methods on brain function, EEG was primarily used to measure brain activity during the working memory task [9,10]. EEG has the advantage of being able to directly and non-invasively measure the electrical signal transmission of the brain during a specific stimulus or task at high temporal resolution [11,12]. However, the acquired EEG signals are vast, and it is quite complicated to process the data. Since the overall analysis of the EEG signals is difficult, studies have been conducted through an analysis limited to certain characteristics presented in the existing literature. This makes it difficult to analyze measurable results objectively, thereby limiting the scope of research to the subjective hypotheses of researchers in the existing literature. For this reason, many recent studies have introduced statistical methods and machine learning techniques to analyze complex EEG measurement signals [13,14,15,16,17].

Machine learning technology is receiving special attention as a method for effectively analyzing EEG data because it has the advantage of finding subtle patterns, even in vast amounts of data measured in various channel signals [9,18,19,20].

In fact, machine learning has been applied in several EEG-based studies, such as EEG classification according to emotion [11,12] and EEG-based emotion recognition [21,22,23]. In addition, this trend is expected to intensify, considering the recent rapid development of machine learning technologies, such as deep learning [24,25,26].

In this study, we applied a machine learning technique to EEG data to distinguish the types of performed breathing by analyzing the changes in the brain oscillatory activities caused by breathing. Therefore, this study aimed to investigate the immediate and direct difference between nose and mouth breathing in brain activities, as well as the effects of additional O_2_ supply to distinguish the types of breathing.

## 2. Materials and Methods

### 2.1. Participants and Experimental Environment

This study was approved by the Institutional Review Board in Gachon University (IRB: 1044396-201706-HR-108-01). All EEG data were obtained from 20 subjects who voluntarily participated and provided written consent (male: 7, female: 13, age: 23.7 ± 2.28 years, non-smokers, right-handed). To minimize the unnecessary effect of the measuring environment on the EEG data and to remove ambient noise, measurements were made in a soundproof room. The EEG electrodes were attached to the scalp in accordance with the 10–20 system. The average temperature of the data measurement environment was 22.6 ± 1.45 °C. The average humidity remained constant at 23.0 ± 2.1%.

### 2.2. Data Acquisition and Preprocessing

Measurements were performed with an EEG (LAXTHA QEEG-32FX) in which we obtained data from 32 channels, electrooculogram (EOG) from 2 channels, and electrocardiogram (ECG) from 1 channel. Using Telescan (LAXTHA), five types of EEG brain wave frequency bands: delta (δ: 0.5–4 Hz), theta (θ: 4–8 Hz), alpha (α: 8–13 Hz), beta (β: 13–30 Hz), and gamma (γ: 30–50 Hz) were sampled at 250 Hz. Data preprocessing was conducted using MATLAB-based EEGLAB (SCCN). The first 1 min out of the total 5 min of epoch measurement data was excluded (Figure 1), and only 4 min of data were used to analyze steady state EEG signals with band-pass filtering (0.5–50 Hz). The EOG affected by visual stimuli and ECG artifacts were removed using the independent component analysis algorithm provided by EEGLAB. The EEG data generated from the above preprocessing underwent a fast Fourier transform analysis. Thus, we obtained the relative spectral density power for the five brain waves (δ, θ, α, β, and γ).

To examine the EEG signal differences in working memory performance during respiration, the EEG signals were measured for 5 min each in the rest, 1-back, and 2-back task states. The subject’s brain waves were measured in two rest states with their eyes closed and open. The n-back task state consisted of a 1-back task and a 2-back task, representing two levels of working memory tasks. In the n-back task state, a number from 0 to 9 was presented to the subject for 0.5 s (Figure 2). Then, a fixation cross (+) was displayed for the next 1.5 s. N-back is a task that requires a response if the currently presented number is the same as the number given n steps before. Therefore, from a working memory perspective, 1-back is a simpler task than 2-back. A multi-parameter patient monitor (MSLMP03) was used to measure the physiological data, namely blood oxygen level (SpO_2_) during both the resting state and the n-back working memory tasks. Also, n-back tasks were performed with the Psytoolkit (www.psytoolkit.org (accessed on 25 February 2021)), and the working memory (WM) accuracy and response time were measured.

During the rest state with closed eyes, the tasks consisted of nose and mouth breathing and mouth breathing with O_2_ supply. In addition, the tasks consisted of the rest state with eyes open and the n-back states during mouth breathing with O_2_ (Figure 3). The O_2_ saturation level of the mixed air used for the O_2_ supply was maintained at approximately 20%. The EEG data for each of the 5-min blocks were collected in a random order with a break of 1 min between measurements for the brain washout of the previous task.

### 2.3. Linear Discriminant Analysis Random Forest (LDARF)

The machine learning algorithm proposed in this paper is a binary classifier applied to EEG data to determine whether the data were obtained from nose breathing or mouth breathing. The algorithm, LDARF, combined a linear discriminant analysis (LDA) with random forest (RF). The EEG data used for learning consisted of 160 attributes (or features), that is, 5 frequencies (δ, θ, α, β, γ) × 32 channels. LDARF extracted effective feature vectors from the EEG data by applying LDA based on supervised learning, and then the machine learning model was trained with the data in the RF. The LDA learned the distribution of EEG data and extracted the feature vectors that produced the optimal decision boundary. The dimension of the input vector was effectively reduced using this process. The next classification was performed by the RF classifier, which is an ensemble technique combining multiple decision trees. This was followed by bootstrap techniques using several decision trees, which allowed the classifier to achieve high generalization ability and minimize over-fitting even with a small amount of learning data. Figure 4 provides a brief representation of the LDARF. The experiment used an RF classification with 1000 decision trees. The decision trees determined each breathing and finally classified the breathing through hard voting.

### 2.4. Training of the LDARF Classifier

For the machine learning training, each EEG was processed as a vector with 160 elements consisting of the relative spectral density power of five brain waves in 32 channels. To train the LDARF classifier to differentiate between mouth and nose breathing through brain waves, the signal recorded at rest with closed eyes was used. By preprocessing the EEG data obtained from the closed eyes rest state, the values of the five brain waves in the 32 channels were converted to a 160-dimensional vector. After the LDARF classifier trained the individual EEG data, it individually determined whether the EEG sample, especially mouth breathing with O_2_ supply, was mouth or nose breathing. In detail, cross-validation (CV) method is utilized for the classifier, in which CV divides the datasets into data folds, for example, 10-fold CV means dividing the entire datasets into 1 test fold and 9 training folds so that the 10 folds can be a test fold and 10 iterations can be performed to train the model. In this study, the repeated 10-fold CV estimator was applied to improve the accuracy of the datasets [27]. That is, 10-fold CV was repeated 50 times to verify the performance of the trained LDARF classifier. A total of 500 LDARF classifiers were generated after training with 500 different training sets through the repeated CV, and they were verified with the test sets which were not seen during the training. Lastly, the state of the test EEG datasets was individually determined regarding mouth or nose breathing, and the results were evaluated with accuracy.

The LDARF classifier was trained using only the EEG signal of closed eyes rest with mouth and nose breathing. Thus, the difference in brain waves between nose breathing and mouth breathing in a stable condition was extracted, and this was used to distinguish between the breathing methods. Figure 5 provides a brief overview of the EEG data used in learning and data used in the inference in the LDARF classifier model.

### 2.5. Identification of EEG Data with Mouth Breathing by the Trained LDARF Classifier

The LDARF classifier could distinguish the signals between mouth and nose breathing in the closed eyes rest condition with high accuracy through learning with steady state EEG signals. Then, the inference was estimated using the EEG data obtained during mouth breathing with O_2_ supply during the four tasks: closed eyes rest, open eyes rest, 1-back, and 2-back. These tasks may have different O_2_ demands according to their difficulty. In 500 LDARF classifiers obtained through fifty 10-fold CVs, inferences to these EEG data were performed. In addition, the accuracy of each classifier was measured.

## 3. Results

The analysis for O_2_ saturation (SpO_2_) and behavioral performance (WM accuracy) were performed for examining the physiological effects according to breathing types (Appendix A). The O_2_ supply via mouth was significantly higher for SpO_2_ than others breathings without additional O_2_ supply (*p* < 0.001, F = 11.118 for 1-back task in ANOVA; see also in the Appendix A for details). However, O_2_ blood saturation of mouth breathing did not differ from that of nose breathing (*p* = 0.945). In addition, the WM accuracy did not differ significantly from each group (*p* = 0.711, F = 0.585 in ANCOVA). Therefore, due to their subtle effects on behavioral performance, machine learning analysis had to be used to investigate whether the breathing types could be detected in brain oscillatory activity.

The performance of the LDARF classifier trained with the EEG data collected during mouth and nose breathing in the closed eyes rest condition was sufficiently high, as shown in Table 1. To determine the possible influence of muscle artifacts, the analysis in the absence of the gamma waves was also conducted, and the results showed that they had no effect on differentiating breathing types (Table 1). The LDARF classifier differentiated the EEG data between the nose and mouth breathing classes with very high accuracy, suggesting that the LDARF classifier successfully extracted the differences of mouth and nose breathing from the EEG signals. Figure 6 shows the accuracy and standard deviation per CV for the 10-fold CV. At 10-fold CV, LDARF showed a high accuracy of 0.984. Figure 7 shows the receiver operating characteristic (ROC) curve and the area under the curve (AUC) of the trained LDARF classifier. The AUC value was 0.991, indicating that the LDARF classifier performed well.

Table 2 shows the results of tasks with different demands of O_2_ supply, which were obtained by applying the previously trained LDARF classifier to the EEG data of those tasks. In Figure 8, the LDARF classifier, which had high accuracy in the closed eyes rest state with mouth and nose breathing, showed a substantially poor performance in mouth breathing with O_2_ supply during the rest with open eyes state. The accuracy of the classifier dropped from 98% for the rest state with closed eyes to 50% for rest with open and closed eyes and mouth breathing with O_2_ supply in the presence of gamma waves. When the LDARF classifier for mouth breathing was trained for O_2_ supply situations, the results were significantly less distinguishable, suggesting that the LDARF classifier was affected by the O_2_-related changes in the EEG signals used for distinguishing between mouth and nose breathing. Oxygen was thought to be more adequately supplied at higher levels of oxygen demand (1-back and 2-back working memory tasks) than in the rest state. These mouth breathings with O_2_ supply were determined to be nose breathing during 67% of the 1-back task and 75% of the 2-back task. Furthermore, when the O_2_ demand increased (from 1-back to 2-back working memory tasks), and sufficient O_2_ was supplied through the mouth, the EEG signal patterns might be more similar to that of nose breathing than mouth breathing. Therefore, the LDARF classifier determined it consistently as nose breathing, even though it was mouth breathing.

The LDARF classifier could quantify which channels and brain waves were more important in distinguishing between mouth and nose breathing with respect to their relative spectral density power while performing dimension reduction through LDA. In terms of machine learning, LDARF could quantitatively calculate the feature importance for 160 input features, as shown in Table 3, which shows the top and bottom five features that were important in distinguishing nose and mouth breathing. The results of Table 3 and Figure 9 show that the gamma brain wave had the greatest influence on the determination of respiratory types from the EEG data. In addition, the ranking of the EEG features (δ, θ, α, β) obtained in the presence of gamma waves was the same as that obtained in the absence of gamma waves (Appendix A).

## 4. Discussion

### 4.1. Confirmation of the Effect of O_2_ Supply during Mouth Breathing through Machine Learning

This study aimed to extract the characteristics of the breathing method from EEG signals using machine learning technology. Furthermore, this study demonstrated the direct and immediate changes in brain function due to lack of O_2_ supplied by mouth breathing. Unlike previous studies limited to the existing otolaryngology treatment for mouth breathing, this study aimed to confirm the necessity of additional O_2_ supply in a work environment that requires extensive brain function and to show that additional O_2_ supply can alleviate or prevent brain function decline caused by mouth breathing. The results showed no differences in behavioral performance depending on the breathing types, but differences in brain activity were clearly detectable through machine learning analysis.

In this study, inferences were performed using the EEG data from the condition with additional supply of O_2_, with different O_2_ demands for different working memory tasks. Each inference was evaluated for accuracy. The complexity level of the inferences for the different states increased in the following order: rest with open eyes, 1-back, and 2-back with mouth breathing with O_2_ supply. Therefore, the O_2_ demand should increase with the level of complexity of the operation; a previous report [10] has concluded that as the cognitive load increases, the brain’s demand for O_2_ increases, and O_2_ supplied in high concentrations is used more efficiently.

This study has shown that machine learning using EEG signals can sufficiently determine the immediate and direct effects of oral and nasal breathing on cognitive function. In addition, mouth breathing affected the working memory performance due to the possible lack of oxygen as the task difficulty increased, but when additional O_2_ was supplied through the mouth, the discriminator, with increasing complexity of the task, increasingly identified mouth breathing as nose breathing.

Moreover, in complex tasks that require more oxygen from the brain, O_2_ supply more effectively reduced the characteristic differences between EEG signals in nose and mouth breathing. These results support the fact that the main difference between the EEG signals in nose and mouth breathing may be due to the extent of O_2_ supply to the brain [1] and that the breathing method has an immediate and direct effect on brain function [2,3,4,5]. Furthermore, machine learning techniques, such as LDA and RF, can be trained with very high performance in EEG signals. In addition, they can be used in EEG studies as an effective analysis method to measure the extent and type of factors (or features) that affect the classification results because they can simplify the measurement of the relative importance of each feature [18,28,29].

A variety of machine learning methods are used in EEG studies: random forests, as well as support vector machines, have been used by researchers to analyze EEG signals and perform dimensionality reduction through principal component analysis [11,14,15,22]. Unlike conventional statistical research methods, machine learning enables inference to determine brain donor status from a single EEG data. Therefore, it can be applied to monitor the state of brain function in an actual work environment.

### 4.2. Side Effects and Various Risks of Mouth Breathing on Working Efficiency

Mouth breathing is very common in adolescents and causes morphological deformation of the face [4,5]. It has also been reported that children breathing through their mouths have significantly lower reading ability and working memory performance in arithmetic than children breathing through their noses [2]. In allergic rhinitis patients, mouth breathing is associated with an increase of approximately 30% in asthma prevalence compared to nose breathing [4]. A high incidence of obstructive sleep disorders in adolescents with mouth breathing has also been reported [6]. Compared to healthy adolescents, adolescents with obstructive sleep apnea syndrome had poorer learning ability, more behavioral problems, and worse concentration ability [2,3]. Similarly, children with respiratory problems during sleep had low language processing and learning ability [3]. As such, various side effects of mouth breathing have been reported, but they were mostly indirect interpretations of the side effects on brain function. Therefore, the aspects of direct and immediate brain functional changes that occur over a long period of time due to mouth breathing also need to be considered.

### 4.3. Additional O_2_ Supply to Avoid Hazards of the Working Environment

The results showed that if additional O_2_ supply is provided during mouth breathing, it is recognized as nose breathing, at least by the brain waves. This means that brain function degradation, such as working memory performance reduction, can be improved through additional O_2_ supply. Prior research has thoroughly investigated differences in brain regions where changes in brain function occur due to breathing. However, there has been no research to show whether the distinction could be made based on the type of breathing. Once machine learning models become more advanced and can distinguish and analyze EEG signals of different breathing conditions effectively, it will be possible to predict the effect of breathing changes on the brain and to prevent breathing habits that damage brain function. Even in the field of cognitive ability, such as working memory tasks related to breathing, the machine learning model can be a very effective approach to analyze EEG signals. This will also allow the field of cognitive science to customize personalized treatment plans and initial interventions for patients based on predictions from EEG signals. Personalized treatment plans can have a significant effect on clinical management transitions [29].

## 5. Conclusions

The present study showed that the management of O_2_ supply could affect brain performance. Similar to this result, other studies have also shown improved work efficiency in working environments where O_2_ supply was increased [7]. An increase in cognitive abilities, such as a decrease in reaction time during the 1-back task, was observed in environments with high-concentration of O_2_ [8]. Oxygen supply affecting brain function will also have a great effect on improving the efficiency in the working environment. However, there were some limitations to this study. First, EEG data with three different levels of O_2_ demand were tested through inference using the LDARF classifier. However, only the data from the condition of mouth breathing with an additional supply of O_2_ were examined, whereas the effect of O_2_ supplied by the nose should also be examined. Further improvements to the generalized performance of LDARF should be made because the data were obtained from a limited number of subjects, although the generalized performance was strictly verified with 50 10-fold CVs. Further research should be conducted to investigate how O_2_ supply will help improve work efficiency in various working environments and why there had no differences in cognitive performance in the current design. In order to draw definite conclusions, participants, such as patients with long-term mouth breathing or cognitive impairment, should be also included in future studies. In addition, the LDA and RF classifiers were used in this study. In future studies, EEG signals need to be analyzed using different types of machine learning models, such as support vector machine, principal component analysis, and independent component analysis. Also, it is necessary to determine which machine learning models are effective in analyzing EEG signals for a specific task. Moreover, considering that the ensemble technique of combining different models in machine learning is superior to one model-based method, the study of the application of the ensemble technique is also of great importance. Furthermore, this study could provide the basis for future research, including the effects of insufficient O_2_ supply on structural brain development and academic performance in children, the effects on cerebral activity while further supplying O_2_ supply to the brain, the effects of gas exchange in the lungs, and/or the respiratory effects of degenerative brain diseases such as vascular dementia, Alzheimer’s, or Parkinson’s disease.

## Figures and Tables

**Figure 1 brainsci-11-00293-f001:**
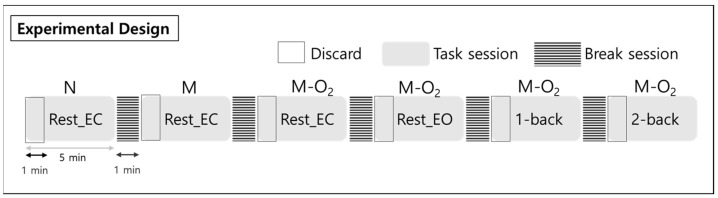
Experimental design for the rest states and n-back tasks, including nose and mouth breathing, and mouth breathing with O_2_ supply. Subjects performed a total of 6 tasks, each a five-minute session, in a random order. There was a break for one minute between tasks, and the data from the first one minute was excluded from the analysis. Note: Rest_EC, resting state with closed eyes; Rest_EO, resting state with open eyes; N, nose breathing; M, mouth breathing; M-O_2_, mouth breathing with O_2_ supply.

**Figure 2 brainsci-11-00293-f002:**
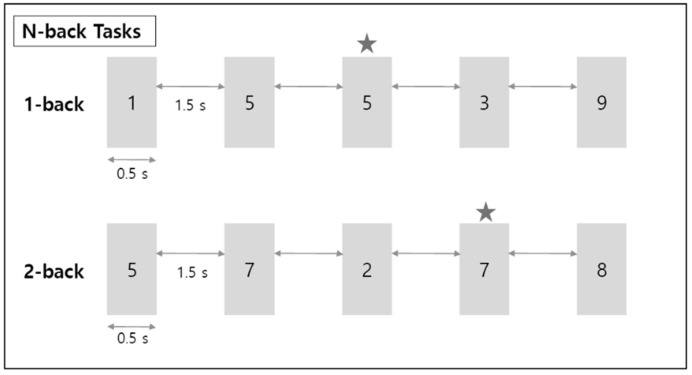
Schematic diagram for n-back working memory task. ★ refers to the target number; when this number appears, the correct button must be pressed.

**Figure 3 brainsci-11-00293-f003:**
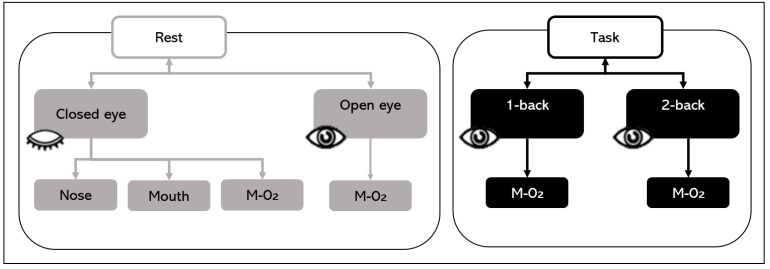
Experimental sessions. In the closed eyes rest state, nose and mouth breathing and mouth breathing with O_2_ supply were tested. Mouth breathing with O_2_ supply tasks (M-O_2_) consisted of rest with closed and open eyes and 1-back and 2-back tasks.

**Figure 4 brainsci-11-00293-f004:**
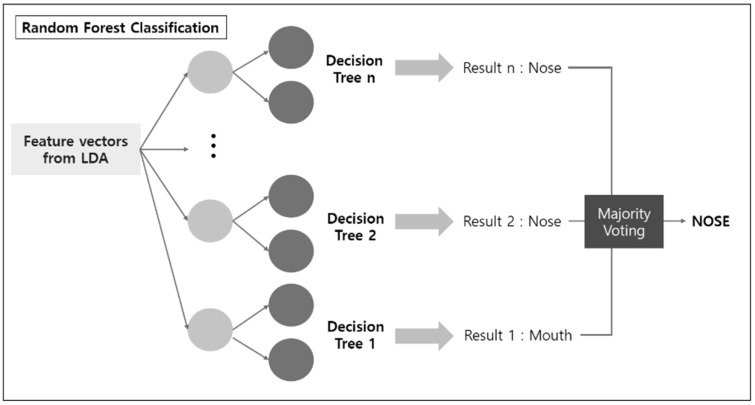
Linear discriminant analysis (LDA) and random forest classification to discriminate electroencephalography (EEG) signals.

**Figure 5 brainsci-11-00293-f005:**
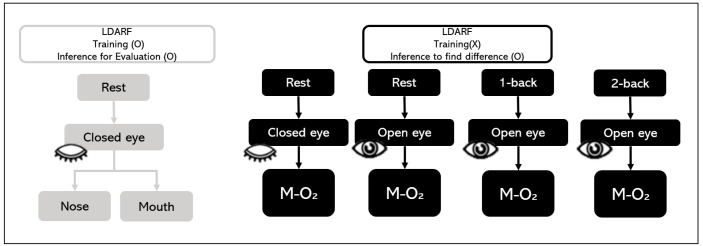
Electroencephalography (EEG) training dataset and EEG test dataset.

**Figure 6 brainsci-11-00293-f006:**
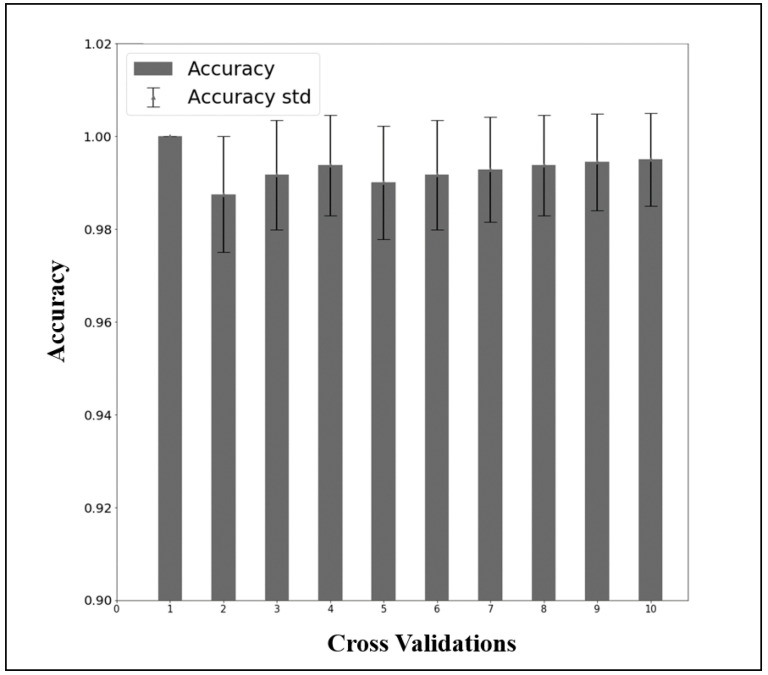
The results of 50 repetitions of 10-fold cross-validation of the data of the rest with closed eyes and mouth and nose breathing.

**Figure 7 brainsci-11-00293-f007:**
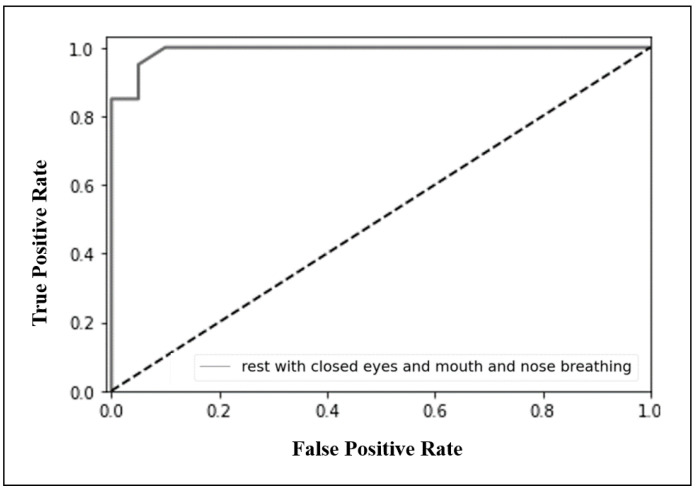
Receiver operating characteristic (ROC) curve and area under the curve (AUC) graph from the condition of rest with closed eyes and mouth and nose breathing, trained with linear discriminant analysis random forest (LDARF) classifier. The AUC of 0.991 confirmed that the LDARF classifier performed well.

**Figure 8 brainsci-11-00293-f008:**
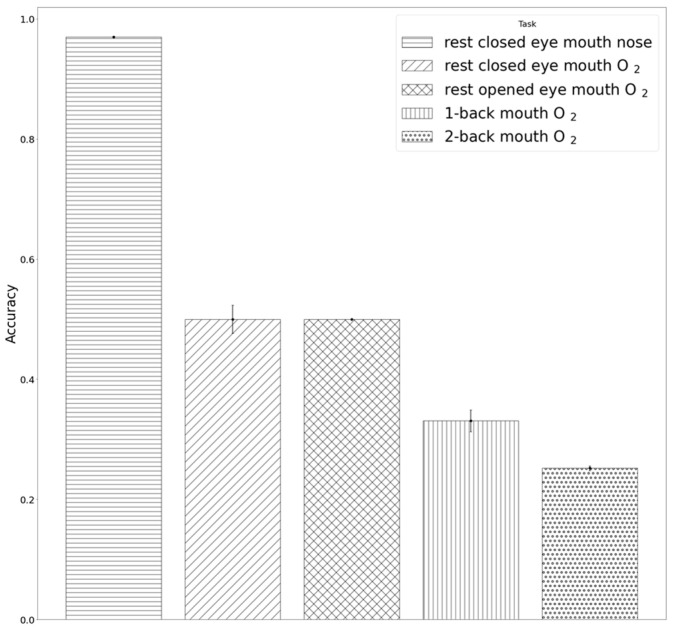
Results of inferences by the linear discriminant analysis random forest (LDARF) classifier. Inference results for electroencephalography (EEG) data of the tasks with three different levels of oxygen demand obtained using 500 LDARF classifier trained with the data recorded during the rest with closed eyes condition.

**Figure 9 brainsci-11-00293-f009:**
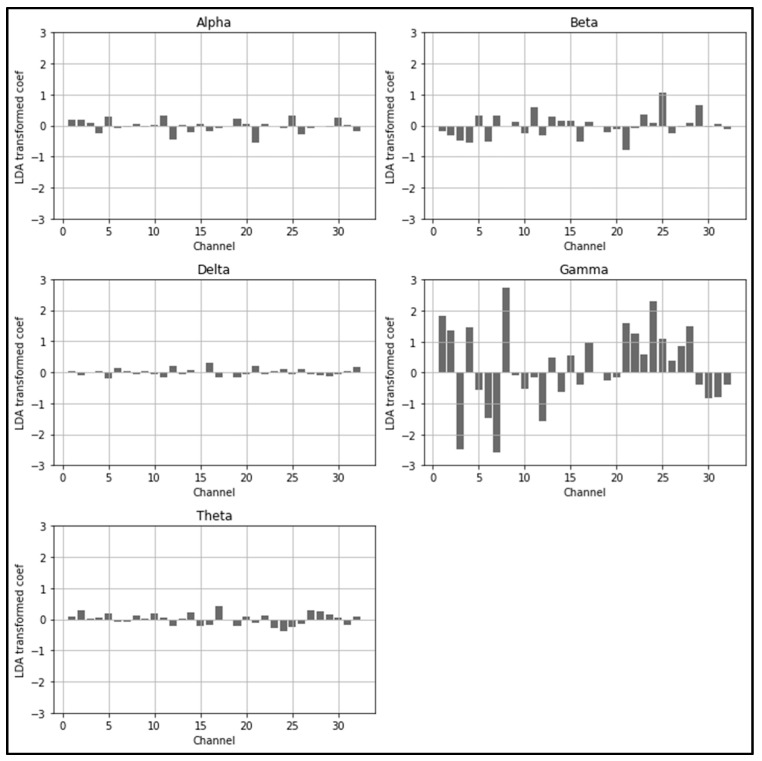
Linear discriminant analysis random forest (LDARF) classifier coefficients for each channel. Gamma brain wave had a significant influence on determining mouth and nose breathing.

**Table 1 brainsci-11-00293-t001:** Results of 50 iterations of 10-fold cross-validation. Inference performance by the linear discriminant analysis random forest (LDARF) classifier with test data subsets (closed eye rest during mouth and nose breathing) in each cross-validation is presented using accuracy, precision, and sensitivity.

	With Gamma Wave	Without Gamma Wave
	Mean ± SD	Mean ± SD
Accuracy	0.984 ± 0.005	0.992 ± 0.001
Precision	0.975 ± 0.011	0.990 ± 0.001
Sensitivity	0.994 ± 0.005	0.995 ± 0.001

Abbreviations: SD, standard deviation.

**Table 2 brainsci-11-00293-t002:** Results of inferences by the linear discriminant analysis random forest (LDARF) classifier during mouth breathing with O_2_ supply. The results of the inferences on the electroencephalography (EEG) data in three different levels of oxygen demand obtained using the LDARF classifier are shown. The classifier was trained with 500 data from closed eyes rest state during mouth breathing. It should be noted that the precision was always equal to one because O_2_ was supplied through mouth and that the sensitivity was the same as the accuracy.

	With Gamma Wave	Without Gamma Wave
Inference	Accuracy ± SD	Accuracy ± SD
Closed eye rest state	0.506 ± 0.024	0.554 ± 0.003
Open eye rest state	0.500 ± 0.000	0.355 ± 0.007
1-back	0.331 ± 0.018	0.354 ± 0.007
2-back	0.252 ± 0.004	0.200 ± 0.006

Abbreviations: SD, standard deviation.

**Table 3 brainsci-11-00293-t003:** The absolute values of the five important upper positive and negative weight vector channels. These channels have a significant influence on the determination of nose or mouth breathing.

	With Gamma Wave		Without Gamma Wave	
	Wave/Location	Absolute Value	Wave/Location	Absolute Value
Upper positive weight vector	Gamma/C3	2.755	Beta/FC6	1.400
	Gamma/T8	2.292	Beta/AF4	1.010
	Gamma/Fp1	1.835	Beta/O1	0.657
	Gamma/CP6	1.600	Beta/AF3	0.646
	Gamma/F8	1.511	Theta/C4	0.630
Upper negative weight vector	Gamma/T7	2.586	Beta/F4	0.571
	Gamma/F7	2.478	Beta/AF3	0.611
	Gamma/P3	1.576	Beta/P3	0.733
	Gamma/FC5	1.454	Theta/F7	1.092
	Gamma/Fp2	0.817	Beta/FC5	1.169

## Data Availability

The data presented in this study are available on request from the corresponding author. The data are not publicly available due to privacy data.

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
