# Peer review of "Identification of Breathing Patterns through EEG Signal Analysis Using Machine Learning"

_brainsci, 2021, doi:10.3390/brainsci11030293_

Round 1

Reviewer 1 Report

The article concerns the scientifically interesting topic of using machine learning in the field of EEG signal analysis. The article describes to investigate the changes in brain function due to lack of oxygen caused by mouth breathing and to suggest a method to alleviate the side effects of mouth breathing on brain function through an additional O2 supply.

My comments to the article:

- I recommend adding a few sentences to Conclusions about plans for the future in this research

- the bibliography should be expanded with more references. Currently, it is narrow for a scientific article. For example, I recommend attaching the publication: Using Neural Networks for Classification of the Changes in the EEG Signal Based on Facial Expressions, Analysis and classification of eeg signals for brain-computer interfaces, Book Series: Studies in Computational Intelligence. As an example of using machine learning on a similar topic from 2020.

Author Response

Reviewer #1:

The article concerns the scientifically interesting topic of using machine learning in the field of EEG signal analysis. The article describes to investigate the changes in brain function due to lack of oxygen caused by mouth breathing and to suggest a method to alleviate the side effects of mouth breathing on brain function through an additional O2 supply.

The authors appreciated your valuable comments. And we carefully revised the manuscript following the reviewer’s comments accordingly.

My comments to the article:

- I recommend adding a few sentences to Conclusions about plans for the future in this research.

R1_1: The following would be important topics in future research plans: 1) in academic aspect, effects of insufficient O2 supply on structural brain development and academic performance in children, 2) in basic science aspects, effects on cerebral activity while further supplying O2 to the brain and effects of gas exchange in the lungs, and/or 3) in clinical aspect, respiratory effects of degenerative brain diseases such as vascular dementia, Alzheimer’s, or Parkinson’s disease.

We included these plans in the main text as suggested by the reviewer. Thank you for your recommendation.

- the bibliography should be expanded with more references. Currently, it is narrow for a scientific article. For example, I recommend attaching the publication: Using Neural Networks for Classification of the Changes in the EEG Signal Based on Facial Expressions, Analysis and classification of eeg signals for brain-computer interfaces, Book Series: Studies in Computational Intelligence. As an example of using machine learning on a similar topic from 2020.

R1_2: We have added more recent and appropriate references according to the reviewer’s suggestion.

Reviewer 2 Report

The study applied machine learning algorithms to classify EEG recording during different types of breathing. The authors state that this approach can provide a tool for recognition of O2 supply to the brain based on the previous findings directly relating breathing with O2 supply and cognitive workload. However, I think that the majority of conclusions of this study are highly overstated:
1) Authors cannot claim that  Line 17 "The results showed that nose breathing guarantees normal O2 supply to the brain, but mouth breathing interrupts the O2 supply to the brain.» In this particular article, the authors did not provide any evidence about changes in O2 saturation, which could, at least, and very indirectly estimate O2 supply to the brain in participants during different types of breathing. The presented results only show a good classification accuracy of EEG recorded during different types of breathing in the resting state. However, there is no direct relationship between ML accuracy rate and O2 supply claimed by authors. 
2) Using the current study design, it is impossible to draw such conclusion that "one of the most important elements distinguishing the effects of mouth and nose breathing on brain function was the difference in O2 supply». The presented weight distribution shows the highest effect on classification accuracy for the gamma band. The gamma-band is usually contaminated by muscle artifacts, which cannot be totally removed by preprocessing steps. The highest weight of gamma spectral density in classification could indicate that recognition of EEG depended rather on residual muscle activity different for the nose and mouth breathing than on O2 supply. To check the possible influence of muscle artifacts, I would suggest the authors apply a more narrow band filter to EEG, for example, 1-25 Hz.
3) The type of breathing does not directly connect sufficient O2 in the environment and brain function. Additionally, even healthy subjects could have substantial differences in the function of external respiration also influencing O2 supply to the brain.  In this case, it is unclear why we need to use time-consuming EEG recordings to detect mouth breathing for "estimation" of O2 supply in daily life when it would be much easy to control sufficient level of O2 in the environment or check O2  blood saturation (Lines 77-78). 
4) The authors also wrote that "this study aimed to investigate the immediate and direct negative effects of the mouth breathing on brain function" (lines 79-81). However, the study itself does not provide any evidence of negative effects such as results of cognitive performance or self-reports. It could be possible if authors would compare behavioral results in WM tasks performed during different breathing types, but these tasks were done only with mouth breathing in 02-mask.  Without these additional data, the decreasing classification accuracy of EEG in these tasks does not exclusively mean the different levels of O2 consumption or negative effects of mouth breathing (Lines 268-264).
5) As I understood, the classifier worked with group data (which is not very clear from the method description). In sense of practical applications, it would be more useful to check recognition of breathing type in EEG at the individual level. In this case, it would be interesting to correlate behavioral performance and classification accuracy. The significant correlations could be associated with the effect of breathing on cognitive abilities. However, the presented findings cannot justify the authors' conclusions (Lines 266, 268 -270, 320).   

Minor issues 
1) missing information about a system of electrode placement (10-20)?
2) names of electrodes should be provided in Table 3, not only numbers

Author Response

Reviewer #2:

The study applied machine learning algorithms to classify EEG recording during different types of breathing. The authors state that this approach can provide a tool for recognition of O2 supply to the brain based on the previous findings directly relating breathing with O2 supply and cognitive workload. However, I think that the majority of conclusions of this study are highly overstated:

1) Authors cannot claim that Line 17 “The results showed that nose breathing guarantees normal O2 supply to the brain, but mouth breathing interrupts the O2 supply to the brain.” In this particular article, the authors did not provide any evidence about changes in O2 saturation, which could, at least, and very indirectly estimate O2 supply to the brain in participants during different types of breathing. The presented results only show a good classification accuracy of EEG recorded during different types of breathing in the resting state. However, there is no direct relationship between ML accuracy rate and O2 supply claimed by authors.

R2_1: To address the concern of the reviewer, we have analyzed peripheral oxygen saturation (SpO2) in this revision to very indirectly estimate O2 supply to the brain. The results showed that the O2 supply via mouth was significantly higher for SpO2 than other breathings (P<0.001, in 1-back task, 2-back task and Resting; see also in the text for details) and the result also showed the amount of O2 was sufficient enough to detect using photoplethysmography (PPG). However, it could not directly reflect the central O2 saturation as the reviewer commented, since SpO2 was measured in PPG using a finger clip type probe. In the further study, therefore, it should be still evaluated in various approaches.

We included these results in the main text (Table S1). Thank you for your recommendation.

2) Using the current study design, it is impossible to draw such conclusion that “one of the most important elements distinguishing the effects of mouth and nose breathing on brain function was the difference in O2 supply.” The presented weight distribution shows the highest effect on classification accuracy for the gamma band. The gamma-band is usually contaminated by muscle artifacts, which cannot be totally removed by preprocessing steps. The highest weight of gamma spectral density in classification could indicate that recognition of EEG depended rather on residual muscle activity different for the nose and mouth breathing than on O2 supply. To check the possible influence of muscle artifacts, I would suggest the authors apply a more narrow band filter to EEG, for example, 1-25 Hz.

R2_2: To examine the effect of gamma wave, we further performed ML analysis in the absence of gamma waves. When comparing with the presence or the absence of gamma waves, the result was similar with that as shown in the previous submitted manuscript. Therefore, muscle artifacts could not have a significant impact on current results, unlike the reviewer’s concern. These results were also included in the main text of this revision (Table 1, 2, 3, and Table S3). Thank you for your valuable opinion.

3) The type of breathing does not directly connect sufficient O2 in the environment and brain function. Additionally, even healthy subjects could have substantial differences in the function of external respiration also influencing O2 supply to the brain. In this case, it is unclear why we need to use time-consuming EEG recordings to detect mouth breathing for "estimation" of O2 supply in daily life when it would be much easy to control sufficient level of O2 in the environment or check O2 blood saturation (Lines 77-78).

R2_3: In SpO2 analysis included in this revision, O2 blood saturation (SpO2) of mouth breathing did not differ from that of nose breathing, that is, SpO2 values cannot distinguish breathing types. Therefore, the present study proposes a new method to distinguish their subtle difference, that is, the EEG machine learning method. In the EEG data, the LDARF classifier model could show the difference, that is, breathing types could be distinguished with an accuracy of more than 95%. Furthermore, from the perspective of LDARF classifier, which was learned to distinguish general mouth breathing and nose breathing with an accuracy of 90% or more, breathing through the mouth when sufficient oxygen was supplied was closer to the EEG characteristics of nose breathing, not mouth breathing.

Authors also agree with the reviewer’s opinion regarding to the time-consuming EEG recordings in daily life. However, this study aimed to investigate the effects of mouth breathing as well as O2 supply via mouth. To do this purpose, we have selected EEG recording as the most convenient method.

4) The authors also wrote that “this study aimed to investigate the immediate and direct negative effects of the mouth breathing on brain function” (lines 79-81). However, the study itself does not provide any evidence of negative effects such as results of cognitive performance or self-reports. It could be possible if authors would compare behavioral results in WM tasks performed during different breathing types, but these tasks were done only with mouth breathing in O2-mask. Without these additional data, the decreasing classification accuracy of EEG in these tasks does not exclusively mean the different levels of O2 consumption or negative effects of mouth breathing (Lines 268-264).

R2_4: As the reviewer suggested, we further analyzed behavioral results such as working memory accuracy (WM accuracy). In the ANCOVA analysis using the control variable as the response time to the stimuli, the WM accuracy did not differ significantly from each other (P=0.711). Therefore, the result suggested that even cognitive performance could not differentiate the breathing types due to their subtle effects on behavioral performance. Note that the group was consisted of 6 working memory tasks, such as 1-back and 2-back mouth, nose and O2 supply via mouth. The result was also included in the main text of this revision (see Table S2).

5) As I understood, the classifier worked with group data (which is not very clear from the method description). In sense of practical applications, it would be more useful to check recognition of breathing type in EEG at the individual level. In this case, it would be interesting to correlate behavioral performance and classification accuracy. The significant correlations could be associated with the effect of breathing on cognitive abilities. However, the presented findings cannot justify the authors' conclusions (Lines 266, 268 -270, 320).

R2_5: Now, the data further provided in this revision may support current conclusions. The difference in the peripheral oxygen saturation (SpO2) level and behavioral performance (WM accuracy) was too subtle to distinguish the breathing types, but the machine learning for EEG data could differentiate them. Furthermore, this developed machine learning algorithm (LDARF classifier) can be applied to the individual levels to determine from which type of breathing EEG data was obtained. The present accuracy over 90% was obtained from individuals who have not been exposed during training.

Minor issues

1) missing information about a system of electrode placement (10-20)?

R2_6: Thank you for your recommendation. We corrected the information in the Methods.

2) names of electrodes should be provided in Table 3, not only numbers.

R2_7: Thank you for your recommendation. We have included the information in the Table.

Round 2

Reviewer 2 Report

The authors responded to all my comments except:

a) The authors did not provide a more clear description of classification (group/individual) in the method section (see commentary 5 in the previous review). 
b) In the discussion and conclusions, the authors keep claiming that the type of breathing affects brain performance. These data have been reported previously, however in this current article: What is brain performance? In results,  the authors added "the WM accuracy did not differ significantly from each group (P=0.711, F=0.585 in ANCOVA). Therefore, the result suggested that even cognitive performance could not differentiate the breathing types due to their subtle effects on behavioral performance." If there were no effect on WM task performance, or it was too subtle, why do we need to determine breathing type using EEG?  Shall authors refer to the previous findings instead of stressing "even cognitive performance". 
I would suggest specifying in the conclusion that breathing type can be detected in brain oscillatory activity using MLA, for me, that is all that the presented results imply.  I would also recommend adding limitations in the discussion sections explaining possible reasons why there were no differences in cognitive performance in the current study design.
